# Autoregressive Motion Generation with Gaussian Mixture-Guided Latent Sampling

**Linnan Tu[1], Lingwei Meng[2], Zongyi Li[1], Hefei Ling[1]\*, Shijuan Huang[1]**
[1]Department of Computer Science and Technology, Huazhong University of Science and Technology
[2]The Chinese University of Hong Kong
{lntu, zongyili, lhefei, shijuan_huang}@hust.edu.cn
lmeng@se.cuhk.edu.hk

## Abstract

Existing efforts in motion synthesis typically utilize either generative transformers with discrete representations or diffusion models with continuous representations. However, the discretization process in generative transformers can introduce motion errors, while the sampling process in diffusion models tends to be slow. In this paper, we propose a novel text-to-motion synthesis method GMMotion that combines a continuous motion representation with an autoregressive model, using the Gaussian mixture model (GMM) to represent the conditional probability distribution. Unlike prior autoregressive approaches relying on residual vector quantization, our model employs continuous motion representations derived from the VAE's latent space. This choice streamlines both the training and the inference processes while mitigating discretization errors. Specifically, we utilize a causal transformer to learn the distributions of continuous motion representations, which are modeled with a learnable Gaussian mixture model. Extensive experiments demonstrate that our model surpasses existing state-of-the-art models in the motion synthesis task.

## 1 Introduction

3D human motion synthesis, *i.e.*, generating a vivid action sequence by control conditions, holds promising applications in game development, embodied intelligence, and the animation. Two main paradigms are used today: (1) One paradigm uses generative transformers with discrete motion representation, such as GPT-like (1; 2; 3) or BERT-like (4; 5) models, to synthesize motions based on specific conditions. Typically, these methods require a vector quantization model (6) to convert continuous motion sequences into discrete codebook tokens. Subsequently, a generative transformer is trained either using a teacher-forcing approach to generate discrete motion tokens autoregressively or using a masked filling strategy to generate them non-autoregressively. Finally, a decoder synthesizes the final motion sequence. (2) The other paradigm employs diffusion models with continuous motion representation (7; 8; 9; 10). They first train a continuous autoencoder, primarily using VAE-based (11) models, to create a compressed and semantically rich representation of motion in latent space. The diffusion model then utilizes various sampling strategies (12; 9; 13) and conditional control methods (14; 15) to generate latent vectors of motion that align with the given conditions. However, both paradigms have disadvantages.

The VQ process inevitably disrupts the continuity of the time series, which can lead to errors during the token connection process. Some works employ residual VQ (RVQ) (4; 16), which involves iteratively summing residuals using multiple codebooks to mitigate compression loss. However, models using RVQ often have a structure similar to VALL-E (17), which requires separate processing

---

\*Corresponding Author

39th Conference on Neural Information Processing Systems (NeurIPS 2025).

of tokens for the primary layer and the residual layers, increasing the complexity of the model. Some studies use binary codebooks (18) or model body joints separately (19), but this can increase the codebook size and lead to higher storage costs. Moreover, the highly compressed nature of the motion can lead to less diverse outputs, favoring common patterns found in the training data.

The diffusion models offer better diversity with text conditions and generate high-quality motion (20; 21; 8); however, their inference speed is limited since the sampling process requires multiple iterations. Efforts to address these issues, such as MotionLCM (12) and B2A-HDM (22), have successfully decreased sampling steps through distillation techniques. Nevertheless, they require pre-specifying a maximum sequence length, which limits the scalability of motion generation. Therefore, we are motivated to develop an approach that reduces compression loss in motion representation while enabling continuous generation in multi-modal spaces.

In this paper, we propose a novel framework called Autoregressive Modeling with Gaussian Mixtures (GMMotion), which aims to synthesize motions with continuous GMM latent sapce, demonstrating that vector quantization is not a necessary prerequisite for autoregressive motion modeling. We lead the motion sequence into the Gaussian mixtures' latent space with learnable parameters in the first stage. By constraining the latent representation to be a continuous multi-modal distribution during VAE training and recovering it in the second stage with a continuous autoregressive (AR) model, we can build a AR model that retains all the advantages of Large Language Modeling (prompting, integrated duration modeling, and sampling) without many of the challenges associated with VQ. Additionally, our key advantage lies in eliminating the need to predefine the duration of generated content, enabling the synthesis of longer motion sequences based on the complexity of control conditions.

Our approach includes three major aspects:

- We utilize a learnable Gaussian mixture model to represent motion sequences as multi-modal distributions.
- We introduce an autoregressive causal transformer that learns the distribution of continuous representations and employs Gaussian mixture sampling to generate motion representations.
- We design a straightforward architecture that benefits from single-step Gaussian mixture sampling and AR generation, leading to extrapolatable inference and high-quality motion synthesis.

## 2   Related Work

### 2.1   Autoregressive Generation with Continuous Tokens

Autoregressive models (23; 24; 17) typically generate content utilizing quantized representations extracted from raw data (6; 25). However, recent studies (26; 27) find that as long as the per-token distributions are modeled, autoregressive models can be approached without vector quantization. MAR (27) introduced an image-masked autoregressive modeling method that uses an MLP head to perform diffusion sampling on several consecutive tokens at each iteration. LatentLM (28) introduces a multi-modal latent language model with a next-token diffusion approach, achieving good results in both speech synthesis and image generation tasks. However, the sequential iterative diffusion sampling process can lead to extremely slow inference speeds. MELLE (29) proposes an autoregressive text-to-speech synthesis model using uni-modal Gaussian sampling to accelerate the inference process. However, as shown in Figure 1, time series data often follow multi-modal distributions, making it difficult to accurately fit with an unimodal Gaussian distribution. VAE (12) exhibits an unimodal distribution but does not cluster the representations of movements. RVQ (4) captures the slightly chaotic multimodal distribution. Due to sampling errors and the varying movements of different joints, the raw data do not conform to a typical normal distribution, resulting in multi-modal distributions or even more complex distributions.

### 2.2   Motion Generation Methods

Research on human motion analysis has a long history (30), statistical models (31; 32; 1) have been employed in earlier studies. Some motion synthesis models leverage raw motion data for training generative models (7). However, these models can be affected by measurement errors in the

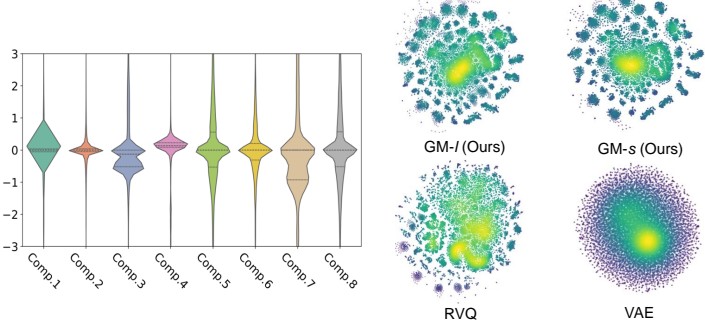

(a) Raw Motion's Distribution    (b) Comparison of Latent Distributions

Figure 1: (a) Violin map of the motion's moving components randomly selected from the normalized HumanML3D datasets;(b) Visualization of different representation model's latent space with t-SNE. Yellow indicates dense sample distribution, while blue represents sparse sample distribution.

data, which often result from inaccuracies in the motion capture process (33; 34). Furthermore, the motion vectors corresponding to different joints exhibit specific distributions, making it challenging for these models to learn representations of complex movements. To maintain the continuity of time series, some approaches (3; 13) employ continuous representation learning models such as variational autoencoders (VAEs) (11; 35). In these models, an encoder predicts the mean and variance of motion latent vectors, followed by sampling from a Gaussian distribution to obtain the latent vector representation, which is then decoded to reconstruct the motion. Although VAEs (8) achieved satisfactory reconstruction results, they struggled to differentiate between various motions, thereby increasing the difficulty of learning for the subsequent generative model.

Recent work has leveraged Vector Quantized-Variational Autoencoders (VQ-VAEs) (5; 36; 18) to achieve discrete representation learning for motion. These models use codebook indices as motion tokens, enabling the application of language modeling (37) techniques and resulting in impressive generation outcomes. However, due to the inevitable quantization error introduced by the discrete process, autoregressive or masked generation methods based on VQ models have been constrained (29; 27).

## 3 Methodology

Given a motion description like "A person rolls forward once, then raises their hand and quickly shoots a ball.", our goal is to create a 3D motion sequence that reflects this description. In this paper, we propose a learnable Gaussian mixture model to represent motion sequences as multi-modal distributions, regularized by the Kolmogorov-Smirnov (KS) distance (Sec. 3.1). We then introduce a masked autoregressive transformer that learns the distribution of continuous representations and employs Gaussian mixture sampling to generate motion representations (Sec. 3.2).

Our proposed model comprises two primary components: a Gaussian Mixture Variational Autoencoder (GM-VAE) and an AR model. The GM-VAE incorporates causal convolutions to preserve temporal consistency in sequential data processing, along with a learnable prior distribution to regularize the latent space. The AR model consists of a text encoder, a causal transformer, a latent sampling module, and a PostNet. The latent sampling module applies Gaussian resampling to the autoregressively generated latent vectors from the transformer to produce coarse reconstructions, which the PostNet then refines into detailed motion outputs.

### 3.1 Stage 1: Learning Continuous Motion Representation

Motion data, which describes the movement of different parts of the body, can be highly variable due to the unique motion patterns of individual joints. This complexity makes it difficult to represent the data with a single normal distribution. For example, the movement of your feet when walking or running (including speed and rhythm) can differ significantly from the movements of other parts of your body. This motivated us to model multi-modal distributions in a continuous latent space to preserve the temporal integrity of motion data (as shown in Figure 1).

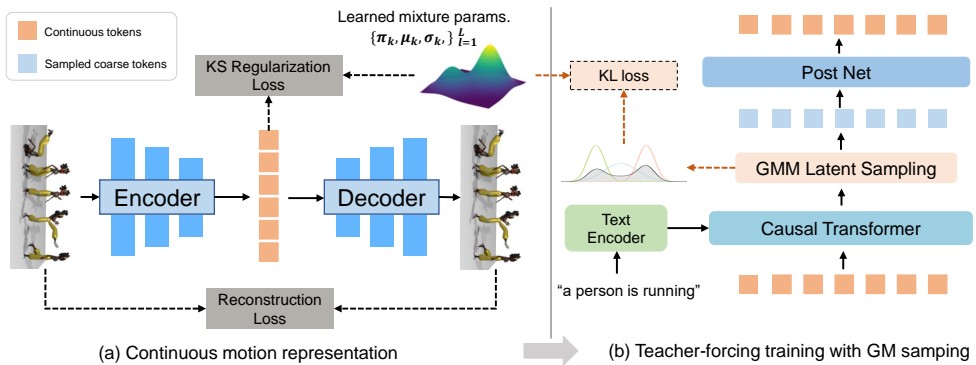

(a) Continuous motion representation | (b) Teacher-forcing training with GM samping

Figure 2: The training stage of the proposed GMMotion model. First, a VAE is trained to reconstruct motion data. The encoder obtains the posterior distribution of motions, which is shaped by a learnable Gaussian mixture, and then the decoder generates reconstructed motions from latent features. Next, a causal transformer generates latent representations, which are regularized by VAE's shared GMM parameters. Text embeddings and latent representations control the transformer to predict Gaussian variables for masked latent. Finally, continuous latent is restored through Gaussian mixture latent sampling.

**Learnable GMM as a prior.** Some works (38; 39) suggest that structured VAEs can effectively train deep models using a GMM as a prior distribution, replacing the normal distribution typically used in vanilla VAEs. Inspired by (39), an unsupervised clustering model using Gaussian mixture variational autoencoders, we propose the learnable mixture of Gaussians as a prior distribution to replace the single normal distribution. The evidence lower bound is:

$$\text{ELBO} = \mathbb{E}_{q(\mathbf{z}|\mathbf{x})}[\log p(\mathbf{x}|\mathbf{z})] - \lambda \text{KL}_{gmm}, \tag{1}$$

where $\lambda$ controls the strength of the regularization, $\mathbb{E}_{q(z|x)}[\log p(x|z)]$ is the log-likelihood of the reconstructed data. Similar to VQ-based models, we obtain the latent vector $z$ from the encoder through a deterministic mapping. However, since the posterior is a deterministic function and the prior is composed of a mixture of Gaussians, directly calculating the difference between these two distributions (known as KL divergence) is not straightforward (40). The reconstruction term, therefore, can be estimated by drawing Monte Carlo samples:

$$
\begin{aligned}
\text{KL}_{gmm} &= \text{KL}(q(\mathbf{z}|\mathbf{x}) \| \sum_{l=1}^{L} \pi_l \mathcal{N}(\boldsymbol{\mu}_l, \boldsymbol{\Sigma}_l)) \\
&\approx \frac{1}{M} \sum_{j=1}^{M} \sum_{l=1}^{L} p_\beta^{(j)} \text{KL}(q(\mathbf{z}|\mathbf{x}) \| p(\mathbf{x}|\boldsymbol{\mu}_l, \boldsymbol{\Sigma}_l, \mathbf{t}_l = 1)),
\end{aligned}
\tag{2}
$$

where $l$ denotes the mixing exponent index, $L$ is the total number of mixtures, and $\mathbf{t}$ is a one-hot vector sampled from the mixing probability $\pi$, which chooses one component from the Gaussian mixture. $M$ is number of samples, $p_\beta^{(j)} = p(\mathbf{t}_l = 1|\mathbf{z}^{(j)})$ is the conditional probability of $\mathbf{t}_l$ being equal to 1 when given the sampled latent $\mathbf{z}^{(j)}$. The gradient can be backpropagated with the standard reparameterization trick (35). The prior term can be calculated analytically.

**KS distance for latent regularization.** Our loss definition is based on the Kolmogorov-Smirnov (KS) test (41) for equality of one-dimensional probability distributions. The KS test serves as a statistical tool to determine whether a set of $N$ one-dimensional data points is drawn from a specified reference distribution. This determination is made by comparing the cumulative distribution function (CDF) of the reference distribution with the empirical CDF $\overline{F}_N$, which is derived from the observed samples.

For each mode $l \leq L$ in the GMM, let $u_l$ represent the mean, $\Sigma_l$ represent the covariance matrix, and $\pi_l$ represent the weight of that specific mode. The CDFs for univariate Gaussian distributions can be defined as:

$$F_{\text{GMM},j}(z) = \sum_{l=1}^{L} \pi_l \Phi\left( \frac{z - [\mu_l]_j}{[\Sigma]_{j,j}} \right), \tag{3}$$

and the covariance matrix of the GMM can be computed as:

$$\Sigma_{\text{GMM}} = \sum_{l=1}^{L} p_l \Sigma_l + \sum_{l=1}^{L} p_l (\mu_l - \bar{\mu})(\mu_l - \bar{\mu})^T, \tag{4}$$

where $\bar{\mu} = \frac{1}{L}\sum_{l=1}^{L} \mu_l$, applying our proposed regularization method to multi-modal GMMs is a simple extension. Given d-dimensional latent samples $z_1, ..., z_N$, the empirical marginal CDF in dimension $j$ is given by:

$$\bar{F}_j^{(N)}(z) = \frac{1}{n} \sum_{n=1}^{N} \mathbb{I}_{[z_n]_j \leq z}, \tag{5}$$

where $\mathbb{I}$ is an indicator function, the primary term in our loss function is specified as:

$$\mathcal{L}_{\text{KS},L}(\mathbf{z}_1, \dots, \mathbf{z}_N) = \frac{1}{d} \sum_{j=1}^{d} \text{MSE}\left( \bar{F}_j^{(N)}(\mathbf{z}_j), F_{\text{GMM},j}(\mathbf{z}_j) \right). \tag{6}$$

Based on a motion-VAE (42), we use an L2 loss between the ground truth poses $\mathbf{x}$ and predictions $\hat{\mathbf{x}}$. We use an L2 loss between the root-centered vertices of the SMPL mesh (43) $\mathbf{v}$ and predictions $\hat{\mathbf{v}}$:

$$\mathcal{L}_{rec}(\hat{\mathbf{x}}, \mathbf{x}) = \|\hat{\mathbf{x}} - \mathbf{x}\|_2^2 + \|\hat{\mathbf{v}} - \mathbf{v}\|_2^2. \tag{7}$$

The final loss function includes motion reconstruction and latent space distribution constraints:

$$\mathcal{L}_{loss} = \lambda_{\text{KS}} \mathcal{L}_{\text{KS},L} + \mathcal{L}_{\text{rec}}. \tag{8}$$

## 3.2 Stage 2: Learning Autoregressive Latent Sampling with GMM

Considering that the motion representation in the first stage is constrained to be continuous and follows a GM distribution, in the second stage, we directly generate this distribution from a variational inference perspective, thereby avoiding the multi-step iterations of diffusion sampling (44).

The model uses a causal transformer architecture like T2M-GPT (36). Additionally, we designed a Residual-MLP composed of three-layer Multilayer Perceptrons to reorganize the sampled latent representations. Then, the coarse latent representations are refined through a PostNet with residual connections to reconstruct finer latent representations. In the case of Gaussian mixture generative modeling, we no longer need an embedding layer or a softmax layer. Since there are no mask/padding token IDs in the continuous case, we design learnable mask latent and padding latent to replace them.

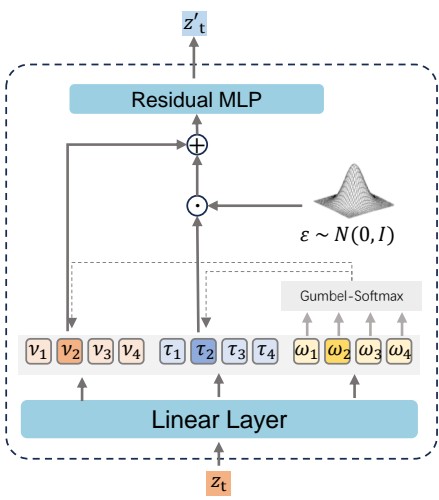

Figure 3: GMM latent sampling process.

**Gaussian mixture latent sampling.** We aim for the transformer to predict the parameters of a mixture of Gaussian distributions, from which we can sample to obtain latent vectors( Figure 3). We define an autoregressive model for the continuous random variable $z_t \in \mathbb{R}^D$, where the conditional probabilities are represented as a mixture of Gaussian distributions:

$$p(\mathbf{z}_t' \mid \mathbf{z}_{t-1}', \dots, \mathbf{z}_1', Y) = \sum_{l=1}^{L} \omega_l^t \mathcal{N}\left( \mathbf{z}_t; \boldsymbol{\nu}_l^t, (\boldsymbol{\tau}_l^t)^2 \right), \tag{9}$$

where $\omega_n^t$ is the n-th GM's weights, $\boldsymbol{\nu}_n^t$ and $\boldsymbol{\tau}_n^t$ represent the n-th GM's mean and diagonal variances for generating step $t$. The mixture parameters are generated by a neural network $f()$, which takes the previous inputs and conditional information as its inputs:

$$[\omega_{1:L}^t, \boldsymbol{\nu}_{1:L}^t, \boldsymbol{\tau}_{1:L}^t] = f(\mathbf{z}_{t-1}, \dots, \mathbf{z}_1, C), \tag{10}$$

where $C$ is the condition embeddings, $z$ is the AR transformer latent representations. We use one Linear layer to obtain the weights, means, and diagonal variances. We use the negative log-likelihood loss function as:

$$\mathcal{L}_{\text{NLL}} = \sum_{i=1}^{M} -\log p\left(\mathbf{z}_i \mid \{\omega_l^i, \nu_l^i, \tau_l^i\}_{l=1}^{L}\right) + \text{KL}(\sum_{l=1}^{L} \omega_l^{1:M} \mathcal{N}\left(\boldsymbol{\nu}_l^{1:M}, \boldsymbol{\tau}_l^{1:M}\right)) \| \sum_{l=1}^{L} \pi_l \mathcal{N}(\boldsymbol{\mu}_l, \boldsymbol{\Sigma}_l)). \tag{11}$$

To ensure the mixture weights and variances are valid, we apply softmax to normalize the $\omega^t$ values and softplus to the $\tau^t$ values in the network output.

**Head pre-padding and learnable rotary position encoding.** Motion sequences in the same batch vary in length. Previous AR motion generation models pad shorter sequences with padding tokens at the tail to maintain uniform length and apply absolute position encoding to indicate positional relationships. Inspired by large language models (45), we fill padding tokens at the head of shorter sequences and employ learnable relative position encoding (46) to preserve positional relationships. Our goal is to enhance the scalability of the autoregressive model, enabling it to synthesize longer motion videos even when trained on shorter sequences from the HumanML3D dataset.

On the other hand, we adopt the same text-conditioning injection method as SALAD (47), where a cross-attention module with residual connections is added after each transformer block to inject text embeddings. We use the same text encoder as LAMP (16). Compared to embedding text conditions as the first token in the AR iteration process, this approach allows for more flexible control and better compatibility with our head pre-filling method.

# 4 Experimental Results

## 4.1 Datasets and evaluation metrics

**Datasets.** To fairly and accurately compare our method with the baseline, we used two main motion-language benchmarks: KITML (34) and HumanML3D (33). The KITML dataset comprises 3,911 actions from KIT motion data, with each action accompanied by one to four text notes (a total of 6,278 notes). The KITML motions are set at 12.5 frames per second (FPS). HumanML3D includes 14,616 actions from the AMASS (48) and HumanAct12 (49) datasets. Each action is described by three text scripts (a total of 44,970 notes). The HumanML3D motions are set at 20 FPS and last up to 10 seconds. We augmented the data by flipping motions and split both datasets into training, testing, and validation sets.

**Evaluation metrics.** We evaluate the generated motions in three aspects: (1) **Quality of the generated motions.** We use the Frechet Inception Distance (FID) to measure how close the generated motion patterns are to the real ones. (2) **Text-motion alignment.** We use the Matching Score to measure how well the texts match the generated motions. Additionally, we apply R-Precision($N$) to assess how accurately motions can be retrieved based on their corresponding texts within a set of $N$ motion-text pairs. (3) **Motion disversity.** MultiModality (MModality) measures the generation diversity conditioned on the same text and Diversity calculates variance through features (33).

## 4.2 Experimental setup

We use the same CNN-based encoder and decoder as Momask (4). We introduce a linear layer after the encoder (the same as (50)) and replace the vector quantization step with a learnable Gaussian mixture distribution. To maintain training stability, we make the mean learnable, initialize the weights with a uniform distribution, and fix the variance to be the identity matrix. The dimension of the 8-layer Causal Transformer is set to 512, with 8 heads and a dropout rate of 0.1, using the GELU activation function. Learnable RoPE embeddings are applied. The diagonal covariance matrices are set to be diagonal.

| Methods | R-Precision ↑ | | | FID ↓ | MM-Dist ↓ | Diversity → | MultiModality ↑ |
| | Top-1 | Top-2 | Top-3 | | | | |
|---|---|---|---|---|---|---|---|
| Real motion | $0.511^{\pm.003}$ | $0.703^{\pm.003}$ | $0.797^{\pm.002}$ | $0.002^{\pm.000}$ | $2.974^{\pm.008}$ | $9.503^{\pm.065}$ | - |
| T2M-GPT (36) | $0.492^{\pm.003}$ | $0.679^{\pm.002}$ | $0.775^{\pm.002}$ | $0.141^{\pm.005}$ | $3.121^{\pm.009}$ | $9.722^{\pm.082}$ | $1.831^{\pm.048}$ |
| AttT2M (51) | $0.499^{\pm.003}$ | $0.690^{\pm.002}$ | $0.786^{\pm.002}$ | $0.112^{\pm.006}$ | $3.038^{\pm.007}$ | $9.700^{\pm.090}$ | $2.452^{\pm.051}$ |
| ParCo (52) | $0.515^{\pm.003}$ | $0.706^{\pm.003}$ | $0.801^{\pm.002}$ | $0.109^{\pm.005}$ | $2.927^{\pm.008}$ | $9.576^{\pm.088}$ | $1.382^{\pm.060}$ |
| MoMask (4) | $0.521^{\pm.002}$ | $0.713^{\pm.002}$ | $0.807^{\pm.002}$ | $0.045^{\pm.002}$ | $2.958^{\pm.008}$ | - | $1.241^{\pm.040}$ |
| MoGenTS (19) | $0.529^{\pm.003}$ | $0.719^{\pm.002}$ | $0.812^{\pm.002}$ | $0.033^{\pm.001}$ | $2.867^{\pm.006}$ | $9.570^{\pm.077}$ | - |
| LaMP (16) | $0.557^{\pm.003}$ | $0.751^{\pm.002}$ | $0.843^{\pm.001}$ | $0.032^{\pm.002}$ | $2.759^{\pm.007}$ | $9.571^{\pm.069}$ | - |
| DiverseMotion (53) | $0.515^{\pm.003}$ | $0.706^{\pm.002}$ | $0.802^{\pm.002}$ | $0.072^{\pm.004}$ | $2.941^{\pm.007}$ | $9.683^{\pm.102}$ | $1.869^{\pm.089}$ |
| MDM (7) | $0.320^{\pm.005}$ | $0.498^{\pm.004}$ | $0.611^{\pm.007}$ | $0.544^{\pm.044}$ | $5.566^{\pm.027}$ | $9.559^{\pm.086}$ | $2.799^{\pm.072}$ |
| MLD (8) | $0.481^{\pm.003}$ | $0.673^{\pm.003}$ | $0.772^{\pm.002}$ | $0.473^{\pm.013}$ | $3.196^{\pm.010}$ | $9.724^{\pm.082}$ | $2.413^{\pm.079}$ |
| MotionDiffuse (54) | $0.491^{\pm.001}$ | $0.681^{\pm.001}$ | $0.782^{\pm.001}$ | $0.630^{\pm.001}$ | $3.113^{\pm.001}$ | $9.410^{\pm.049}$ | $1.553^{\pm.042}$ |
| ReMoDiffuse (55) | $0.510^{\pm.005}$ | $0.698^{\pm.006}$ | $0.795^{\pm.004}$ | $0.103^{\pm.004}$ | $2.974^{\pm.016}$ | $9.018^{\pm.075}$ | $1.795^{\pm.043}$ |
| Fg-T2M++ (56) | $0.513^{\pm.002}$ | $0.702^{\pm.002}$ | $0.801^{\pm.003}$ | $0.089^{\pm.004}$ | $2.925^{\pm.007}$ | $9.223^{\pm.114}$ | $2.625^{\pm.084}$ |
| GMMotion (Ours) | $0.572^{\pm.003}$ | $0.761^{\pm.003}$ | $0.852^{\pm.001}$ | $0.086^{\pm.003}$ | $2.743^{\pm.008}$ | $9.792^{\pm.085}$ | $2.033^{\pm.058}$ |
| Real motion | $0.424^{\pm.005}$ | $0.649^{\pm.006}$ | $0.779^{\pm.006}$ | $0.031^{\pm.004}$ | $2.788^{\pm.012}$ | $11.08^{\pm.097}$ | - |
| T2M-GPT (36) | $0.416^{\pm.006}$ | $0.627^{\pm.006}$ | $0.745^{\pm.006}$ | $0.514^{\pm.029}$ | $3.007^{\pm.023}$ | $10.92^{\pm.108}$ | $1.570^{\pm.039}$ |
| AttT2M (51) | $0.413^{\pm.006}$ | $0.632^{\pm.006}$ | $0.751^{\pm.006}$ | $0.870^{\pm.039}$ | $3.039^{\pm.021}$ | $10.96^{\pm.123}$ | $2.281^{\pm.047}$ |
| ParCo (52) | $0.430^{\pm.004}$ | $0.649^{\pm.007}$ | $0.772^{\pm.006}$ | $0.453^{\pm.027}$ | $2.820^{\pm.028}$ | $10.95^{\pm.094}$ | $1.245^{\pm.022}$ |
| MoMask (4) | $0.433^{\pm.007}$ | $0.656^{\pm.005}$ | $0.781^{\pm.005}$ | $0.204^{\pm.011}$ | $2.779^{\pm.022}$ | - | $1.131^{\pm.043}$ |
| DiverseMotion (53) | $0.416^{\pm.005}$ | $0.637^{\pm.008}$ | $0.760^{\pm.011}$ | $0.468^{\pm.098}$ | $2.892^{\pm.041}$ | $10.87^{\pm.101}$ | $2.062^{\pm.079}$ |
| MoGenTS (19) | $0.445^{\pm.006}$ | $0.671^{\pm.006}$ | $0.797^{\pm.005}$ | $0.143^{\pm.004}$ | $2.711^{\pm.024}$ | $10.92^{\pm.090}$ | - |
| LaMP (16) | $0.479^{\pm.006}$ | $0.691^{\pm.005}$ | $0.826^{\pm.005}$ | $0.141^{\pm.013}$ | $2.704^{\pm.018}$ | $10.93^{\pm.101}$ | - |
| MDM (7) | $0.164^{\pm.004}$ | $0.291^{\pm.004}$ | $0.396^{\pm.004}$ | $0.497^{\pm.021}$ | $9.191^{\pm.022}$ | $10.85^{\pm.109}$ | $1.907^{\pm.214}$ |
| MLD (8) | $0.390^{\pm.008}$ | $0.609^{\pm.008}$ | $0.734^{\pm.007}$ | $0.404^{\pm.027}$ | $3.204^{\pm.027}$ | $10.80^{\pm.117}$ | $2.192^{\pm.071}$ |
| MotionDiffuse (54) | $0.417^{\pm.004}$ | $0.621^{\pm.004}$ | $0.739^{\pm.004}$ | $1.954^{\pm.062}$ | $2.958^{\pm.005}$ | $11.10^{\pm.143}$ | $0.730^{\pm.013}$ |
| ReMoDiffuse (55) | $0.427^{\pm.014}$ | $0.641^{\pm.004}$ | $0.765^{\pm.055}$ | $0.155^{\pm.006}$ | $2.814^{\pm.012}$ | $10.80^{\pm.105}$ | $1.239^{\pm.028}$ |
| Fg-T2M++ (56) | $0.442^{\pm.006}$ | $0.657^{\pm.005}$ | $0.781^{\pm.004}$ | $0.135^{\pm.004}$ | $2.696^{\pm.011}$ | $10.99^{\pm.105}$ | $1.255^{\pm.078}$ |
| GMMotion (Ours) | $0.481^{\pm.005}$ | $0.703^{\pm.006}$ | $0.819^{\pm.004}$ | $0.198^{\pm.012}$ | $2.604^{\pm.023}$ | $11.12^{\pm.095}$ | $1.457^{\pm.039}$ |

Table 1: Quantitative evaluation results on the test sets of HumanML3D (top) and KIT-ML (bottom). ↑ and ↓ denote that higher and lower values are better, respectively, while → denotes that the values closer to the real motion are better. Red and blue colors indicate the best and the second best results.

## 4.3 Motion representation performance

In Table 2, we compared our motion GM-VAE with other motion tokenizers, such as RVQ-VAE (4), Transformer-VAE (8), and VQ-VAE (36), and found that our model demonstrates superior results in motion reconstruction.

| Methods | FID ↓ | MPJPE ↓ | R-Precision ↑ | | |
| | | | Top 1 | Top 2 | Top 3 |
|---|---|---|---|---|---|
| VQ-VAE | $0.081^{\pm.001}$ | $72.6^{\pm.001}$ | $0.483^{\pm.003}$ | $0.680^{\pm.003}$ | $0.780^{\pm.002}$ |
| RVQ-VAE | $0.029^{\pm.001}$ | $31.5^{\pm.001}$ | $0.497^{\pm.002}$ | $0.693^{\pm.003}$ | $0.791^{\pm.002}$ |
| Trans-VAE | $0.023^{\pm.001}$ | $13.7^{\pm.001}$ | $0.499^{\pm.002}$ | $0.695^{\pm.003}$ | $0.791^{\pm.003}$ |
| GM-VAE-$s$ (Ours) | $0.004^{\pm.001}$ | $9.4^{\pm.001}$ | $0.514^{\pm.002}$ | $0.703^{\pm.002}$ | $0.819^{\pm.003}$ |
| GM-VAE-$l$ (Ours) | $0.008^{\pm.001}$ | $10.2^{\pm.001}$ | $0.518^{\pm.002}$ | $0.710^{\pm.002}$ | $0.811^{\pm.002}$ |

Table 2: **Reconstruction results** of latent encoders in our method vs baseline methods(VQ-VAE (36), RVQ (4) and VAE (12)) on HumanML3D (33) data. $s$ and $l$ mean 128 dims and 512 dims in GMM latent spaces.

We also analyzed how our latent space representation compares with others, as shown in the t-SNE plot (see Figure 1). Constrained by the standard normal distribution, VAE (12) exhibits an unimodal distribution but does not cluster the representations of movements. Meanwhile, due to uneven utilization of the codebook, RVQ (4) captures representations of high-frequency movements but is less sensitive to low-frequency movements. Our model not only represents multi-modal motion distributions more effectively but also achieves better clustering of motion data. This is facilitated by the use of Gaussian mixture distributions, which allows the model to capture detailed motion characteristics through unsupervised clustering.

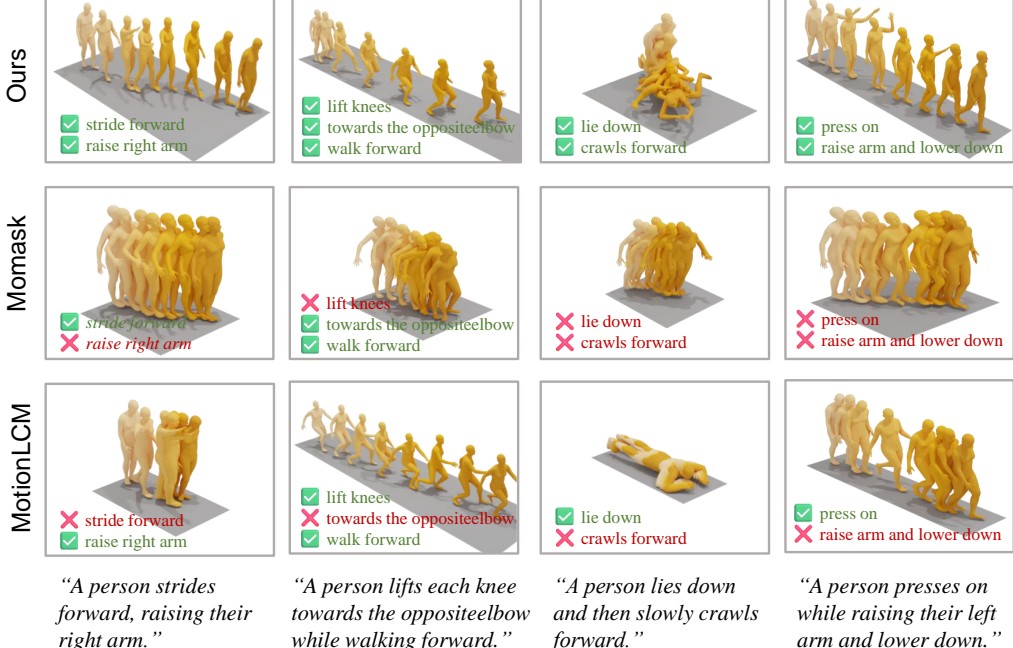

Figure 5: Visualization of qualitative results vs. diffusion based model (12) and VQ based model (4). The color from light yellow to dark yellow indicates the motion sequence order.

## 4.4 Text to motion Generation

**Quantitative results.** We compared our model with other state-of-the-art methods (5; 4; 36; 12; 7), which can be broadly divided into two types: VQ-based models and diffusion-based models. The results indicate that GMMotion achieves favorable outcomes in both text alignment and motion quality. In the qualitative analysis, we evaluated the quality of motion synthesis, the alignment between text and motion, and the diversity of motions. For the quantitative analysis, we compared our model's motion synthesis results with those of other baselines using the same text instructions.

Additionally, to demonstrate the feasibility of applying our method, we compare average inference time results in Appendix.

**Qualitative results.** We also visualize our qualitative results in Figure 5. GMMotion demonstrates more accurate and natural-looking motions compared to the other models. For instance, in the action of striding forward with a raised arm, our model captures the movement fluidly, whereas Momask and MotionLCM exhibit some blurriness and less precise limb positioning. Similarly, in the knee-lifting motion, our model shows clearer and more realistic leg movements, while the other models struggle with the finer details. Overall, our model outperforms Momask and MotionLCM in generating coherent and lifelike human movements across different actions.

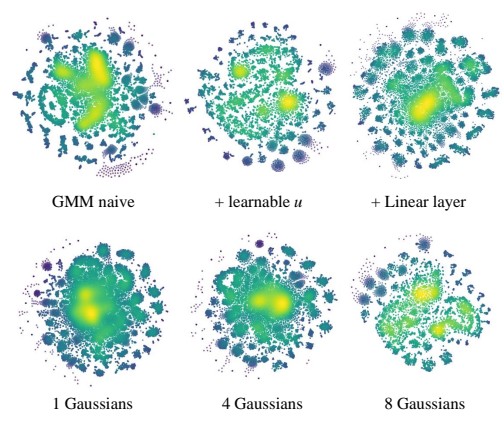

Figure 4: Visualization of different GMM settings with t-SNE.

## 4.5 Ablation Study

We focused on two aspects: (1) The effectiveness of the GMM. We explored its reconstruction performance under various conditions, including the number of components and regularization weights. (2) The continuous motion Transformer structure. We investigated how the transformer module, designed for continuous sampling, influences generation outcomes. We also discuss the

effectiveness of the generation methods in the Appendix. We examined the combined effects of different sampling and training approaches.

**Ablation of Gaussian mixtures.** In Table 3, we examine how the number of components and the loss weight $\lambda$ influence the generative performance. It is evident that the reconstruction quality improves significantly when the number of components exceeds one, suggesting that a Gaussian mixture distribution has better fitting capabilities than a single Gaussian. We also adjusted the weight $\lambda$ of the regularization constraint in the GM-VAE, and the results indicate that the model performs best when the weight is set to 1. In Figure 4, we find that adding a linear layer in the vanilla VAE and setting the GMM parameters to be learnable can improve the latent space representation.

| No.Gaus | KS weight $\lambda$ | FID↓ | Matching score↓ | R-Pre.↑ |
|---|---|---|---|---|
| 1 | 0.1 | $0.121^{\pm.004}$ | $2.939^{\pm.008}$ | $0.815^{\pm.006}$ |
| | 1.0 | $0.145^{\pm.005}$ | $2.942^{\pm.007}$ | $0.805^{\pm.005}$ |
| | 10.0 | $0.165^{\pm.004}$ | $3.080^{\pm.006}$ | $0.799^{\pm.004}$ |
| 4 | 0.1 | $0.088^{\pm.003}$ | $2.781^{\pm.006}$ | $0.841^{\pm.004}$ |
| | 1.0 | $0.086^{\pm.003}$ | $2.743^{\pm.008}$ | $0.852^{\pm.001}$ |
| | 10.0 | $0.142^{\pm.004}$ | $2.892^{\pm.007}$ | $0.827^{\pm.005}$ |
| 8 | 0.1 | $0.101^{\pm.002}$ | $3.011^{\pm.011}$ | $0.801^{\pm.004}$ |
| | 1.0 | $0.196^{\pm.009}$ | $3.110^{\pm.009}$ | $0.782^{\pm.004}$ |
| | 10.0 | $0.659^{\pm.016}$ | $3.556^{\pm.010}$ | $0.679^{\pm.006}$ |

Table 3: Text-to-motion results with different Gaussian components and KS weights.

**Ablation of AR architecture.** In Table 4, we present the results of ablation studies on the structure of the AR model. The findings indicate that the head pre-padding and RoPE modules improve the quality of motion, while the motion reorganization module has a significant impact on the synthesis effect. We believe the motion reorganization module plays a crucial role in refining the rough latent representations after sampling. When we removed Gaussian

| Components | FID↓ | Matching score↓ | R-Pre.↑ |
|---|---|---|---|
| Full | $0.086^{\pm.003}$ | $2.743^{\pm.008}$ | $0.852^{\pm.001}$ |
| *w/o RoPE* | $0.133^{\pm.006}$ | $2.851^{\pm.010}$ | $0.823^{\pm.007}$ |
| *w/o Res* | $0.945^{\pm.028}$ | $3.423^{\pm.018}$ | $0.752^{\pm.009}$ |
| *w/o GM Samp.* | $0.485^{\pm.013}$ | $3.241^{\pm.014}$ | $0.791^{\pm.004}$ |
| *w/o Post* | $0.141^{\pm.004}$ | $2.939^{\pm.010}$ | $0.813^{\pm.005}$ |

Table 4: Results of AR. Where *Res* is the residual net, *GM Samp.* is the Gaussian mixture sampling stage, and *Post* is the PostNet.

sampling, reverting the model to deterministic sampling, there was a notable decline in performance. This suggests that the stochastic nature of Gaussian sampling is essential for maintaining high-quality motion synthesis.

**User study.** To evaluate the perceptual quality of text-driven motion generation, we conducted a user study with 19 participants comparing our method against two baselines: LaMP (16), which generates motions by discrete AR models; and MotionLCM (50), which generates motions by continuous diffusion models. For each

| Methods | Visual Quality | Text-motion Alignment |
|---|---|---|
| LaMP (16) | $3.962^{\pm.171}$ | $3.775^{\pm.186}$ |
| MotionLCM (50) | $3.418^{\pm.163}$ | $3.219^{\pm.196}$ |
| GMMotion (Ours) | $4.392^{\pm.093}$ | $4.121^{\pm.098}$ |

Table 5: User study results.

method, participants were presented with 15 video examples and asked to evaluate them based on two criteria: visual quality and text-motion alignment. All ratings were collected using a 5-point Likert scale ranging from 1 (poorest) to 5 (best). The results demonstrate that our model exhibits advantages in terms of visual quality and text-motion alignment.

## 5 Conclusion

We introduced GMMotion, a novel text-to-motion synthesis framework that employs continuous motion representation and GMM to capture multimodal human motions. GMMotion streamlines training and inference by avoiding vector quantization, instead sampling from learnable GMMs in the latent space. Our two-stage model—first modeling motion sequences into multimodal distributions with GMM, then using a causal transformer for efficient generation—outperforms existing models in quality and alignment.

**Limitations.** While GMMotion achieves efficient motion synthesis through autoregressive generation, the sequential nature of the process may occasionally lead to minor error accumulation over long sequences. Early prediction inaccuracies (e.g., subtle joint angle deviations) could propagate temporally, potentially affecting the smoothness of extended motions.

**Acknowledgments**

This work was supported in part by the Natural Science Foundation of China under Grant 62372203 and 62302186, in part by the Major Scientific and Technological Project of Shenzhen (202316021), in part by the National key research and development program of China(2022YFB2601802), in part by the Major Scientific and Technological Project of Hubei Province (2022BAA046, 2022BAA042).

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
