# OpenReview forum: "Autoregressive Motion Generation with Gaussian Mixture-Guided Latent Sampling"
_NeurIPS.cc/2025/Conference — NeurIPS 2025 poster_

### Official Review · Reviewer_eYpb · 2025-06-29

**Clarity:** 2
**Significance:** 3
**Originality:** 3
**Rating:** 4
**Confidence:** 4

**Summary:**

This paper proposes a novel text-to-motion method GMMotion combines the continuous motion representation (GM-VAE) and autoregressive generation model.
The method is trained in two stages:
1. GM-VAE encodes motion sequences into a multimodal latent space regularized by GMM and KS distance to match real data distribution.
2. Autoregressive Transformer predicts GMM parameters and samples latent vectors directly without diffusion or discrete token decoding.
Such approach avoids vector quantization, which causes a discretization errors and breaks temporal continuity.

**Questions:**

1. Inference time
The autoregressive approach could take long time generating frame-by-frame, presenting the inference time could provide better understanding on the proposed method.
2. Long sequence
Does the proposed method generalizes to the long sequences?

**Ethical Concerns:**

["NO or VERY MINOR ethics concerns only"]

**Final Justification:**

Since the concerns are well addressed by rebuttal, I retain my original recommendation score.

**Limitations:**

Yes.

**Paper Formatting Concerns:**

No.

**Quality:**

3

**Strengths And Weaknesses:**

Strength
1. Novel approach
Authors proposed novel approach to learn the latent space of the motion with GMM.
2. Good generation quality
Qualitative and quantitative results are good.

Weakness
1. Unclear comparison over previous ideas
Since the previous observation on VAE indicates that the generation results using VAE are blurry, authors could provide more detailed explanation why their proposed methods achieved the better performance compared to VAE and VQ-VAE.

---

> ### Author Rebuttal · Authors · 2025-07-29
>
> We thank the reviewer eYpb for their insightful comments and suggestions.
>
> We acknowledge the issues and weaknesses raised by the reviewer. Given that our focus is on text to motion synthesis, explaining inference time is indeed crucial. Furthermore, we will further clarify the advantages of GM-VAE and update this in the paper.
>
> ---
>
> **[W1]**: *Unclear comparison over previous ideas about GM-VAE.*
> *   The blurry results generated by VAEs are commonly observed in image synthesis. We believe this occurs because the KL divergence encourages the latent space distribution to be close to a standard normal distribution, which may cause latent samples to cluster too tightly, resulting in generated images lacking diversity and appearing blurry. In the context of human motion reconstruction studied in this paper, this might manifest as **missing fine-grained motion details between frames** (e.g., subtle hand joint movements or slight head shakes failing to be reconstructed accurately).
> *   Compared to a standard VAE, we define a learnable Gaussian Mixture Model (GMM) prior to prevent the latent space distribution from collapsing into a single, restrictive mode. Under the same model architecture, when training a VAE with a fixed standard normal prior, we obtain an FID of 0.040. In contrast, our GM-VAE achieves a significantly lower FID of 0.008. We will include these comparative results in the appendix.
> *   Compared to VQ-VAE, our method synthesizes motion continuously over time, avoiding motion errors introduced by the discretization process. Since motion sequences are continuous time-series data, VQ-VAE typically discretizes every frame into a single ID. This approach can lead to discontinuities in the reconstructed motion (e.g., MoMask [1] with RVQ achieves an FID of 0.029).
> *   GMM better reconstructs the spatial distribution of human motion. Unlike images and audio, each joint of the human skeleton follows specific motion patterns (which we visualize in Figure 1.a as a **multimodal distribution**), whereas the KL divergence constraint in VAE assumes a **unimodal distribution**, making it difficult to fully reconstruct motion details. The visualization results of the latent space are shown in Figure 1.b.
>
> ---
>
> **[Q1]**: *The inference time should be presented.*
>
> - We evaluate inference time using the average inference time per sentence (AITS) over 100 samples on a single Nvidia 4090 device. Our model's AITS is 0.033, which outperforms MotionLCMv2 [2] (0.045) and MoMask (0.046). More detailed results are available in Appendix D.3.
> - Although the autoregressive (AR) process requires frame-by-frame synthesis, in the first stage we condense every four motion frames into one latent frame. This design gives our model an advantage in parameter efficiency compared to the diffusion-based transformer MotionLCMv2 and MoMask, which requires two separate transformers. MoMask first uses a base transformer for masked generation to obtain base tokens and then employs a residual transformer for non-autoregressive, multi-step iterative synthesis to obtain residual tokens, introducing additional computational overhead. MotionLCM [2] achieves its best results after distillation using 4 iterative steps; our model achieves competitive results with single-step sampling while having significantly fewer parameters (MotionLCM can also use 1-step sampling, but with a noticeable drop in synthesis quality).
>
> - We will clearly state the inference time in the main text of the revised manuscript.
>
> ---
>
> **[Q2]**: *Does the proposed method generalize to long sequences?*
>
> - Yes. We appreciate the reviewer notes this special design and future direction. Some video demonstrations of long-sequence synthesis were attached at the end of the abstract when the main paper was submitted. Currently, the visual quality of long-sequence synthesis still requires refinement due to rendering problem.
>
> - In this paper, we introduce two additional designs to facilitate long-sequence synthesis: prefix padding and rotary position encoding. Since motion sub-sequences within a batch often have varying lengths, autoregressive generation methods like T2M-GPT [3] in the motion generation domain typically pad sequences at the end (suffix padding) for training convenience. However, this approach limits long-sequence synthesis because the model stops inference as soon as it generates a specific stop token.
> Inspired by LLM inference designs, we opt for prefix padding to align the end of motion sequences, combined with rotary positional encoding instead of absolute positional encoding. This design reduces the model's strong dependence on absolute positional relationships, thereby enhancing its scalability.
>
> - We will include a discussion section in the new version elaborating on the capability for long-sequence synthesis. Additionally, we will provide qualitative results of long-sequence synthesis in the appendix.
>
> ----
> We sincerely appreciate the reviewer's constructive suggestions and believe that the additional analysis and explanations improve the quality of our submission. We hope that this provides sufficient reasons to raise the score.
>
> ----
> ### **Reference**
>
> [1] Guo C, Mu Y, Javed M G, et al. Momask: Generative masked modeling of 3d human motions. CVPR 2024.
>
> [2] Dai W, Chen L H, Wang J, et al. Motionlcm: Real-time controllable motion generation via latent consistency model. ECCV 2024.
>
> [3] Zhang J, Zhang Y, Cun X, et al. Generating human motion from textual descriptions with discrete representations. CVPR 2023.

---

### Official Review · Reviewer_Nig7 · 2025-07-01

**Clarity:** 4
**Significance:** 4
**Originality:** 4
**Rating:** 5
**Confidence:** 3

**Summary:**

To address the motion errors introduced by the discretization process in generative transformers as well as the slow sampling of diffusion models, this paper presents a text-to-motion synthesis method GMMotion which aims to achieve autoregressive motion generation and leverage Gaussian mixture model for the representation of the conditional probability distribution. The paper shows comprehensive evaluations and the proposed method achieves SOTA performance.

**Questions:**

1. Please provide the results of training and inference times to fully verify the proposed method.
2. Please explain why the proposed method has relatively low motion disversity (i.e., MultiModality).

**Ethical Concerns:**

["NO or VERY MINOR ethics concerns only"]

**Final Justification:**

After reading the author rebuttal, my cncerns have been addressed. So I keep my score (Accept).

**Limitations:**

Yes

**Paper Formatting Concerns:**

No paper formatting concerns.

**Quality:**

4

**Strengths And Weaknesses:**

Strengths:
1. The paper is well-motivated and significant. The motion errors in generative transformer and the slow sampling of diffusion model are two significant and valuable issues in text-to-motion synthesis.
2. The proposed method is quite novel. It leverages VAE’s latent space and Gaussian mixture sampling for continuous motion representations to reduce the discretization error.
3. The authors conduct comprehensive experimental evaluations and ablation studies, and effectively demonstrate the superiority of the proposed method.

Weaknesses:
1. Lack of time comparisons in the experiments. As one of the motivation is the slow sampling in diffusion model, the authors should provide the results of training and inference times to fully verify the proposed method.

---

> ### Author Rebuttal · Authors · 2025-07-29
>
> We thank the reviewer Nig7 for their careful and constructive review.
>
> We agree with the reviewer's suggestion. Since one of our motivations is to improve the sampling speed of diffusion models, it is important to provide inference speed metrics in the experimental results section of the main text. Additionally, we will further supplement the training details, including the type of GPU used for training and the training duration, to enhance the completeness of our experiments.
>
> -----
>
> **[W1/Q1]**: *Provide the results of training and inference times to fully verify the proposed method.*
>
> *   **Training Time:**
>     We use 4 NVIDIA RTX 4090s to complete the experiments. The training of the GM-VAE takes approximately 14.5 hours. Specifically, the first stage (training the GM-VAE) takes about 14.5 hours, and the second stage (training the AR transformer) takes about 13.6 hours.
>     According to our experiments, in the first-stage training (for VAE, VQ-VAE [1], etc.), our speed is comparable to discrete motion representations (MoMask [2], ~ 16 hours) and continuous motion representations (MLD [3], ~ 15 hours). In the second-stage training, our speed is faster than MoMask ( ~ 24 hours) and MLD (2-3 days). MoMask requires training two separate transformers (base and residual, total parameters ~ 44.9M), while MLD's skip transformer has more parameters (~ 30M) and its diffusion denoising validation process is time-consuming. Our model only requires training a single transformer (~ 11.6M), and its validation process allows for single-step sampling (vs. MLD's DDIM=50), resulting in faster training.
>
> *   **Inference Time:**
>     We use the average inference time per sentence (AITS) over 100 samples on a single Nvidia 4090 device to evaluate inference time. Our model's AITS is 0.033, which is better than MotionLCMv2 [4] (0.045) and MoMask (0.046). More detailed results can be found in Appendix D.3. We will indicate the AITS for different models in the next version.
>
> ----
>
> **[Q2]**: *Explain why the proposed method has relatively low motion diversity (i.e., MultiModality).*
>
> We think this reduction in diversity may stem from the random fluctuations introduced by the Gaussian Mixture parameters. During training, the diagonal values of the covariance matrix are fixed to 1, meaning each Gaussian component is orthogonal. This design helps the model learn the motion distribution more effectively, but at the cost of some diversity.
>
> MultiModality measures the diversity of generated motions within each textual description [5]. On the HumanML3D [5] dataset, our model's MultiModality score (2.033) is still better than existing SOTA models (MoMask with 1.241, MotionLCMv2 with 1.758), but lags behind some classic models (MDM [6] with 2.799, MLD with 2.413). Our model also shows the degree of performance degradation in diversity.
>
> Balancing the quality (e.g., FID) and diversity (e.g., MultiModality) of motion synthesis is a problem worth our consideration. GMM, through uncertainty modeling, struggles to achieve high diversity while maintaining good motion quality, and we will continue to work further in this direction.
>
> ----
> ### **Reference**
>
> [1] Yuan W, Shen W, et al. Motion Generation based on Spatial-Temporal Joint Modeling. Neurips 2024.
>
> [2] Guo C, Mu Y, Javed M G, et al. Momask: Generative masked modeling of 3d human motions. CVPR 2024.
>
> [3] Chen X, Jiang B, et al. Executing your Commands via Motion Diffusion in Latent Space. CVPR 2023.
>
> [4] Dai W, Chen L H, Wang J, et al. Motionlcm: Real-time controllable motion generation via latent consistency model. ECCV 2024.
>
> [5] Guo C, et al. Generating Diverse and Natural 3D Human Motions From Text. CVPR 2022.
>
> [6] Tevet G, et al. Human Motion Diffusion Model. ICLR 2023.

---

> > ### Comment · Reviewer_Nig7 · 2025-08-05
> > **Official Comment by Reviewer Nig7**
> >
> > Thank you for the detailed response. After reading the author rebuttal, my cncerns have been addressed.

---

> > > ### Author Response · Authors · 2025-08-06
> > >
> > > Thank you for your review and response! Your feedback is valuable for us to further improve our work. We appreciate your support.

---

> > > > ### Comment · Reviewer_kPaJ · 2025-08-07
> > > >
> > > > Thanks for author's explaination. The rebuttal address most of my concerns. I raise the score to borderline accept.

---

### Official Review · Reviewer_kPaJ · 2025-07-03

**Clarity:** 3
**Significance:** 2
**Originality:** 2
**Rating:** 4
**Confidence:** 5

**Summary:**

This paper proposes a new text-to-motion generation framework, which incorporates GMM latent space and Auto-regressive modeling to achieve AR generation scheme without discrete tokenization.

**Questions:**

Please kindly refer to the weaknesses part.

**Ethical Concerns:**

["NO or VERY MINOR ethics concerns only"]

**Final Justification:**

The authors' rebuttal has addressed most of my concern. Therefore I vote for borderline accept.

**Limitations:**

Limitations have been well discussed.

**Quality:**

3

**Strengths And Weaknesses:**

Strengths

1. This paper includes several valuable ablation studies, which is beneficial for the community.
2. The implementation details are described in a clear and systematic manner, making the overall behavior of the proposed method easy to follow and reproduce.

Weaknesses

My primary concerns lie in two areas: the novelty of the proposed generative approach and the overall generation quality.

1. From the perspective of novelty, the use of a GMM latent space combined with an autoregressive (AR) decoder is not new, and has already been explored in other domains such as video generation (e.g., DiCoDe[1]) and speech synthesis (e.g., GMM-LM[2]). The current method appears to largely transfer these ideas to the text-to-motion domain without incorporating domain-specific adaptations or insights that account for the unique structure or semantics of motion data.
2. Regarding performance, while the authors claim that discrete token representations introduce significant quantization errors, the proposed method does not show a clear advantage in generation quality over existing approaches. In particular, it underperforms both discrete-token-based methods such as MoMask and its successors, and continuous representation methods like MotionLCM-v2, in terms of overall motion realism and diversity. Although the proposed model achieves the best R-Precision, I would caution against overemphasizing this metric as a primary indicator of generation quality. R-Precision values surpassing the ground truth may suggest that the generated results closely resemble samples from the training set, which may reflect mode memorization rather than true semantic alignment or diversity. Moreover, the metric can disproportionately benefit generations that are statistically similar to frequent training patterns, making it less reliable for evaluating generalization or creativity.
3. Additionally, the claim in Appendix D.1 that AR exhibits strong generative capacity appears to conflict with results in Table 6, where the AR variant shows no clear advantage over MAR. In fact, when the latent dimension is set to 128, MAR significantly outperforms AR, which undermines the claim of AR’s superiority.

[1] Li et al. DiCoDe: Diffusion-Compressed Deep Tokens for Autoregressive Video Generation with Language Models
[2] Lin et al. Continuous Autoregressive Modeling with Stochastic Monotonic Alignment for Speech Synthesis

---

> ### Author Rebuttal · Authors · 2025-07-28
>
> We would like to thank the reviewer for their thoughtful comments.
>
> ---
>
> **[W1]** :  *The novelty of the proposed generative approach. there have been similar works in the fields of video generation and speech synthesis.*
>
> Our motivation for choosing autoregressive motion synthesis within the GMM latent space is as follows: In the domain of human motion, the movement characteristics of a specific joint often exhibit a multimodal distribution (as shown in Figure 1.a, in the violin plot, it is clearly visible that human motion follows a multi-modal distribution, unlike images and audio.). GMMs are inherently well-suited for modeling such multimodal distributions. This is different from the motivation of DiCoDe and GMM-LM. We recognize and appreciate these two excellent works—GMM-LM has already been cited in this paper, and we will include related work in the updated version.
> - In DiCoDe, the Gaussian Mixture Model is used to model the uncertainty of deep tokens in the continuous value space and learns the variance during the autoregressive process. **It still requires a diffusion model to complete the video generation.**
> - In GMM-LM, it estimates the KL divergence through Monte Carlo sampling, whereas our approach adopts the KS distance for regularization.
> - In our method, our autoencoding process keeps the time sequence of human motion and uses Gaussian mixture models for spatial relationships, specifically designed for text to human motion generation.
>
> On the other hand, we agree that it’s valuable to adapt and further design outstanding works from similar domains to fit the characteristics of motion:
>
> - T2M-GPT [1] (text to motion) draws inspiration from VQGAN [2] (image generation), utilizing VQ-VAE + Generative Transformer to achieve autoregressive motion synthesis.
> - Momask [3] (text to motion) references VALL-E [4] (audio synthesis), employing RVQ with two transformers to model motion features at both the base layer and residual layer, achieving excellent generation results.
>
> - MotionLCM [5] (text to motion) takes cues from the application of consistency models [6] in image generation, designing a one-step sampling method for motion diffusion models, which significantly accelerates the sampling speed.
>
> Thus, there have been many successful cases where excellent algorithms from external domains (image generation, speech synthesis, video generation) were adapted to achieve good results. To the best of our knowledge, we are the first to introduce GMM sampling into the motion generation field. We have also been exploring whether new methods can replace diffusion models or VQ-VAE. In this work, we initiated such an exploration.
>
> ---
>
> **[W2]** :  *The overall generation quality.*
>
> - *It's not just R-Precision， our model surpasses existing sota models across many metrics.*
>
> Compared to discrete-token-based methods such as MoMask [3], our model shows clear gains in R-Precision↑ (0.852 vs 0.807), MM-Dist↓ (2.743 vs 2.958), and MultiModality↑ (2.033 vs 1.241). When compared to continuous representation methods like MotionLCMv2 [5], our model also demonstrates significant improvements in R-Precision↑ (0.852 vs 0.837), MM-Dist↓ (2.743 vs 2.773), and MultiModality↑ (2.033 vs 1.758). Furthermore, our model leads in most metrics when compared to T2M-GPT [1], another autoregressive generative model. In the first-stage motion representation, RVQ used by Momask, etc. (FID=0.029) performs worse in motion reconstruction than our method (FID=0.008), which further supports that continuous motion representation achieves better reconstruction.
>
> - *R-Precision still remains a meaningful reference for evaluating text-motion alignment.*
>
> The calculation of R-Precision involves mixing the text description corresponding to the target motion with 31 other negative text samples from the test set, then using a text feature extractor trained from HumanML3D [7] to obtain text vectors for matching with the target motion. As noted in MoMask and MARDM [8], this feature extractor has been in use for almost four years, and it is indeed possible for the R-Precision value to exceed that of the ground truth.
>
> The controversy surrounding R-Precision arises more from the relatively outdated feature extractor used for evaluation. We acknowledge your concern regarding potential mode memorization due to certain historical reasons. To address this, MARDM [8] introduced the HumanML3D dataset with non-redundant data and trained a new feature extractor. Results on this new dataset further validate the overall generation quality of our model.
>
> ### Table 1 Text to motion results on MARDM's new HumanML3D benchmark
>
> | Methods | R-precision | FID       | Matching score | MModality |
> | ------- | ----------- | --------- | -------------- | --------- |
> | Momask  | 0.786       | 0.116     | 3.353          | 1.263     |
> | MARDM   | 0.795       | 0.114     | 3.270          | 2.231     |
> | Ours    | **0.799**   | **0.103** | **3.259**      | **2.427** |
>
> On the new dataset proposed by MARDM, the evaluator has been retrained to better suit current state-of-the-art models. Under this dataset, our method outperforms both Momask and MARDM.
>
> ---
>
> **[W3]**: *Advantages of the AR approach over the MAR approach*
>
> - *With all other hyperparameters being equal, our model (AR) has only half the number of parameters compared to the MAR approach.*
>
> The specific MAR structure we refer to in Appendix D.1 follows MARDM, which consists of a Transformer autoencoder and a Transformer autodecoder, implementing masking and pseudo-reordering between them.
>
> - *The benchmark used in the appendix is the MARDM-HumanML3D with redundant parameters removed, rather than the original HumanML3D.*
>
> MARDM argues that a large portion of the human motion parameters in the original HumanML3D are not effectively utilized (about over 60%), and thus removes these redundant features, further reducing data abundance. This reduction may make it more difficult for small-parameter models to learn complex motion synthesis.
>
> Therefore, we believe the reason the AR approach underperforms when the dimension (dim) is set to 128 is due to the model having too few parameters, combined with lower data abundance in this specific dataset, making it difficult to learn motion patterns. Additionally, the AR approach performs better than MAR when the dim is set to 512. We see strong potential in the AR approach for long-motion synthesis and continuous motion generation, and believe it holds future promise for integration with motion LLM-style models.
>
> -----
>
> Overall, we hope this paper can inspire new thinking about text-to-motion synthesis, potentially paving the way for future approaches that could replace either continuous diffusion or discrete AR generation. One key reason we explored the AR approach is to align motion synthesis models with the autoregressive frameworks common in large language models. Additionally, through designs like rotary position encoding, we aim to preserve the ability to generate long sequences, hoping to offer a novel method for motion synthesis.
>
> We hope our response improves your perception of this work, and welcome any further questions or discussions.
>
> ---
> ### **Reference**
> [1] Zhang J, Zhang Y, Cun X, et al. Generating human motion from textual descriptions with discrete representations. CVPR 2023.
>
> [2] Esser P, Rombach R, Ommer B. Taming transformers for high-resolution image synthesis. CVPR. 2021.
>
> [3] Guo C, Mu Y, Javed M G, et al. Momask: Generative masked modeling of 3d human motions. CVPR 2024.
>
> [4] Wang C, Chen S, Wu Y, et al. Neural codec language models are zero-shot text to speech synthesizers.arXiv:2301.02111.
>
> [5] Dai W, Chen L H, Wang J, et al. Motionlcm: Real-time controllable motion generation via latent consistency model. ECCV 2024.
>
> [6] Luo S, Tan Y, Huang L, et al. Latent consistency models: Synthesizing high-resolution images with few-step inference. arXiv:2310.04378.
>
> [7] Guo C, et al. Generating Diverse and Natural 3D Human Motions From Text. CVPR 2022.
>
> [8] Meng Z, Xie Y, Peng X, et al. Rethinking diffusion for text-driven human motion generation. CVPR 2025.

---

> > ### Author Response · Authors · 2025-08-06
> >
> > We sincerely appreciate your careful review of our work. From your response, we can see that you have thoroughly read both our main text and supplementary materials, and we are truly grateful for this.
> >
> > As the discussion period draws to a close, please do not hesitate to contact us if you have any further questions.
> >
> > Overall, we clarify that we did not simply transfer models from other domains (such as GMM in video generation or audio synthesis) directly to our task. Instead, we carefully considered the data distribution characteristics specific to the motion generation field. To the best of our knowledge, we are the first to apply GMM to text-to-motion generation, incorporating specific modifications tailored to this domain. Given the novelty of our approach, we acknowledge that our method underperforms the latest models on certain metrics (e.g., FID). We recognize the excellent performance of works such as Momask and MotionLCMv2, and we hope to incorporate their network designs and training strategies in the future to further enhance the generation quality of our GMM-based approach.
> >
> > If our responses have adequately addressed your concerns, we hope this can serve as a favorable factor in improving our evaluation score.

---

### Official Review · Reviewer_QPHE · 2025-07-03

**Clarity:** 4
**Significance:** 2
**Originality:** 3
**Rating:** 5
**Confidence:** 4

**Summary:**

The paper addresses the challenge of text-to-motion generation and introduces GMMotion, a novel framework that combines a Gaussian Mixture Model (GMM)-based motion compression VAE with an autoregressive transformer. This design circumvents limitations in prior approaches: unlike VQ-VAEs, which introduce discretization errors, the proposed continuous latent representation avoids quantization altogether; and compared to standard VAEs with unimodal Gaussian priors, the use of a GMM better captures the inherently multimodal nature of human motion.

GMMotion operates in two stages: (1) learning a continuous, structured latent space through a GMM-based VAE (GM-VAE) regularized by the Kolmogorov-Smirnov distance, and (2) training an autoregressive causal transformer that models these latent representations and samples from the learned GMM to generate motion sequences. This approach enables prompt-conditioned generation without predefining output length.

Empirical results show the superiority of the method: the GM-VAE outperforms alternative motion tokenizers (including VQ-VAEs and RVQ-VAEs) in reconstruction metrics, and the full generative pipeline produces higher-quality, better-aligned, and more diverse motions across multiple automatic metrics (e.g., FID, R-Precision) and human evaluations. The model also demonstrates faster inference than diffusion-based methods, making it both effective and efficient.

**Questions:**

Q1: The method advocates for using a GMM prior in the VAE. Could you elaborate on why the task of human motion generation specifically benefits from such a prior, as opposed to other generative modeling domains like image, video, or audio generation, where the use of a GMM prior in VAEs is relatively uncommon? These tasks similarly have a multi-modal nature (e.g. different kinds of audio - speech, music, ambient noises).

Q2: In many generative models that use a VAE as a pre-processing stage, the predicted variances tend to collapse to very small values (often due to low KL loss weights), effectively making the VAE behave like an autoencoder. In your case, the diagonal variances are fixed to 1. Doesn’t this introduce significant noise into the latent representations? Alternatively, do the mean values compensate by increasing in magnitude? It would be helpful to clarify how this design choice affects the latent space behavior and model stability. Please also provide some analysis about the magnitude of the predicted means.

Q3: Table 3 shows that the model performs best with only 4 Gaussian components, while using 8 components leads to worse performance than a single Gaussian. How do you explain this? Intuitively, I would expect a more expressive multimodal prior to benefit from a larger number of components. Additionally, could you report performance with 2 components? It would also be helpful to include statistics about the learned GMM (e.g., entropy of the component weights or effective number of components used) to better understand how the model utilizes the mixture.

**Ethical Concerns:**

["NO or VERY MINOR ethics concerns only"]

**Final Justification:**

After reading the authors' rebuttal and the other reviews, I opt for keeping my initial "accept" rating.

**Limitations:**

There is some discussion about the limitations, in section 5.

**Paper Formatting Concerns:**

The formatting seems to be ok.

**Quality:**

3

**Strengths And Weaknesses:**

**Strengths**

Clarity: The paper is very well structured, clear, and easy to follow. In my opinion, there is a good balance between motivating the problem, providing the required background, formulating the solution, describing the experiments, and presenting a thorough ablation study.

Quality: The method appears to be technically sound, and the empirical evidence convincingly demonstrates its success in improving the quality of text-to-motion generation. The ablation study in Section 4.5 further contributes to understanding the role of different components in the proposed method.

Originality: To the best of my knowledge, the idea of combining a continuous VAE with a GMM prior in the context of an autoregressive model for motion prediction is novel.

**Weakness**

I have some questions and concerns about the proposed method, which may affect its overall significance (and also caused me to reduce my Quality rating for the paper). These concerns focus on how effectively the Gaussian Mixture Model (GMM) prior is utilized in the method. Specifically:
- Table 3 shows that the best performance is achieved with only 4 components, and using 8 components actually performs worse than a single Gaussian. This raises concerns about whether the GMM prior is being fully leveraged. It is unclear if this reflects a limitation in training stability, an issue of overfitting, or whether the added expressivity of larger mixtures does not translate to meaningful gains in this setting.
- The covariance matrices are fixed as identity matrices (i.e. all diagonal variances are 1).

Please see expanded questions in the section below.

---

> ### Author Rebuttal · Authors · 2025-07-29
>
> We thank reviewer QPHE for the constructive suggestions and thorough review.
>
> The reviewer has provided detailed thoughts on our GMM design and raised many interesting questions. We acknowledge and appreciate the reviewer's inquiries and feedback, and will include corresponding content and discussions in the revised version.
>
> ---
>
> **[Q1]**: *Why is a GMM prior beneficial for human motion generation in VAEs?*
>
> - *Human motion sequences are continuous time series with specific multimodal distributions.* The representation pattern of human motion is similar to the Mel-spectrogram in speech representation, but each dimension has a specific meaning (e.g., the relative coordinates of the left elbow joint, the angular velocity of the right knee joint). We observed that the motion of each dimension has specific patterns and constraints (e.g., the human forehead cannot rotate 360 degrees). Therefore, unlike sound synthesis and image synthesis, human motion representation has a unique motion distribution. After visualization, we found that it resembles a mixture of Gaussian distributions.
>
>  - *We aim to avoid learning motion patterns through the diffusion denoising process.* Current state-of-the-art methods use diffusion models for motion synthesis and have achieved good results. Many of these methods draw inspiration from image generation and speech synthesis, sometimes overlooking the temporal characteristics and spatial regularities of human motion. We believe treating human motion as an image or a noise sequence introduces redundant steps (for example, motion diffusion models require multiple sampling steps, which may introduce additional computational overhead).
>
>   - *We believe that GMM also holds potential in speech and image synthesis, it is suitable for modeling continuous tokens.* In the first stage of training, a VAE is typically trained for content compression. In practice, the KL divergence loss weight is often constrained to a very small value, making the VAE function more like an autoencoder, where the KL loss primarily serves to limit the numerical range of the latent vectors. In the second stage, diffusion models are frequently employed to learn the noise distribution, mainly because the original time series and images do not exhibit clear distributional patterns. Inspired by MAR [1], an increasing number of approaches （MELLE [2], GMM-LM [4], DiCoDe [3]） are attempting to impose strong constraints on the latent space and then perform sequence synthesis through a Gaussian sampling head. Therefore, we believe that Gaussian Mixture Models remain a promising direction worth exploring.
>
> ---
>
> **[W2/Q2]**: *The VAE's diagonal variances are fixed to 1,  doesn't this introduce significant noise into the latents? Or do the mean values compensate by growing larger?*
>
>  - *The diagonal variances are fixed to 1 to provide uncertainty.* Similar to your observation, when we make the variances learnable, the model exhibits a "laziness" behavior, causing all variances to converge towards 0. We share the same perspective as DiCoDe: even though the text-to-motion task is deterministic, we can explicitly introduce variance into the target. Introducing variance allows for better capture of the uncertainty inherent in the deep latent distribution, which is crucial for generation diversity.
>
>  - *The mean magnitudes of different components vary to adapt to specific motion distributions.* In our reparameterization design, each dimension in the latent space corresponds to an independent GMM. Taking our 128-dimensional model as an example, we analyze the GMMs at dim 1, 32, and 64, as shown in the table below. It can be observed that while the magnitudes of some Gaussian components' means increase, the majority fluctuate around 0.
>
> | position | Number of Gaussians | Mean Value |
> | --- | --- | --- |
> | dim=1 | 1 | [0.1628] |
> | dim=32 | 1 | [-0.1259] |
> | dim=64 | 1 | [0.1924] |
> | dim=1 | 4 | [-0.3429, 0.7895, -0.0567, 0.9218] |
> | dim=32 | 4 | [0.4462, -0.1294, 1.6289, -0.8481] |
> | dim=64 | 4 | [-0.2350, 0.5608, 0.1413, -0.4146] |
> | dim=1 | 8 | [0.3429, -0.7695, 1.5345, 0.0467, -0.9218, 0.6989, -0.1334, 0.4762] |
> | dim=32 | 8 | [0.0042, -1.0907, 0.0902, -0.2431, -0.2554, 0.2300, 0.4215, -0.9123] |
> | dim=64 | 8 | [-2.6504, 0.1296, -0.0042, -0.8976, 0.7694, -3.4231, 0.0008, 0.0321] |
>
> ---
>
> **[W1/Q3]**: *Why does performance degrade with more components, contrary to the expectation that greater expressiveness helps? Could you also report results with 2 components and include GMM usage statistics?*
>
> - *More number of Gaussian mixture components can lead to more uncertainty.* Since we use GMM as a learnable prior (while VAEs generally use a Gaussian distribution as a non-learnable prior), a larger number of Gaussian components allows latent vectors more freedom to select different components, which weakens the constraining power of the GMM prior. More detailed results (we add GMM with 2, 6 components) are shown in the table below. We will include visualizations of the utilization of different GMM components in the next version to further illustrate the above points.
>
> - *GMM can be somewhat analogous to the MoE structure in LLMs, where an excessive number of components leads to increased model complexity.* As the components number increases, the proportion of parameters allocated to each component decreases, and the learning difficulty increases. Our method assigns a Gaussian component for reparameterization based on the probability proportional to the weight during the sampling process, a design similar to the routing mechanism and expert models in MoE. An excessive number of components would result in an increase in model parameters and computational load, making training more unstable. Considering this, we plan to supplement the training loss curves in the next versions.
>
>
> | Number of Gaussians | FID ↓ | Matching score ↓ | R-precision ↑ |
> | ------------------- | ----- | ---------------- | ------------- |
> | 1                   | 0.145 | 2.942            | 0.805         |
> | 2                   | 0.097 | 2.801            | 0.829         |
> | 4                   | 0.086 | 2.743            | 0.852         |
> | 6                   | 0.139 | 2.861            | 0.815         |
> | 8                   | 0.196 | 3.110            | 0.782         |
>
> - We select the same textual instruction to perform inference when the number of components is 4 or 8, and observe the weights corresponding to the dim at 0. The results of GMM weight are
>
> components=2 : [0.5423, 0.4577],
>
> components=4 : [0.2849, 0.2163, 0.2733, 0.2255],
>
> components=8 : [0.1146 0.1348, 0.1053, 0.1069 0.1379, 0.1199, 0.1494, 0.1312].
>
> Here, the results are obtained by printing the GMM sampling head parameters. Although somewhat incomplete, they indicate that the GMM weight distribution is roughly uniform. In future versions, we plan to provide results with higher visualization quality.
>
> ---
> Reviewer prompted us to further explore GMM's potential and limitations in human motion generation. We found that too many components can degrade performance, and fixing variance to 1 requires balancing uncertainty and training stability. We will add discussion and further experiments in the next version.
>
> ---
> ### Reference
>
> [1] Li T, Tian Y, et al. Autoregressive Image Generation without Vector Quantization. Neurips 2024.
>
> [2] Meng L, et al. Autoregressive Speech Synthesis without Vector Quantizatio. ACL 2025.
>
> [3] Li et al. DiCoDe: Diffusion-Compressed Deep Tokens for Autoregressive Video Generation with Language Models.
>
> [4] Lin et al. Continuous Autoregressive Modeling with Stochastic Monotonic Alignment for Speech Synthesis. ICLR 2025.

---

> > ### Comment · Reviewer_QPHE · 2025-08-01
> >
> > I want to thank the authors for their detailed answers and additional analysis.

---

> > > ### Author Response · Authors · 2025-08-06
> > >
> > > Thank you for your review and response! Your comments are very helpful for us to further improve our work. We appreciate your recognition.

---

### Decision · Program_Chairs · 2025-09-17

**Decision:**

Accept (poster)

**Comment:**

This paper proposes GMMotion, a text-to-motion synthesis framework that integrates a Gaussian Mixture VAE (GM-VAE) with an autoregressive transformer, enabling continuous latent motion representation without discrete tokenization or diffusion-based sampling. The method addresses key issues of quantization error in VQ-based models and slow inference in diffusion models. Across evaluations, it demonstrates competitive or superior performance in reconstruction, generation quality, efficiency, and alignment with text prompts.

Three reviewers (QPHE, Nig7, eYpb) are clearly supportive, highlighting novelty, technical soundness, and thorough empirical evaluation. One reviewer (kPaJ) initially questioned the novelty and generation quality but, after the rebuttal and clarifications, raised their score to borderline accept. Overall consensus is positive.

Strengths identified:
1) Novel combination of GM-VAE latent representation and AR transformer for text-to-motion.
2) Clear improvements in reconstruction quality and faster inference compared to diffusion baselines.


Remaining weaknesses and revision requests for the camera-ready version:
1) Clarify novelty relative to prior GMM-based generative models in other domains (e.g., video, speech) and better articulate domain-specific contributions for motion synthesis.
2) Inference and Efficiency: Include explicit runtime comparisons (training and inference) against baselines, as requested by multiple reviewers.
3) Generation Diversity: Discuss the slightly lower diversity compared to some baselines and outline potential improvements.
4) Clarity: Improve explanations on why GM-VAE avoids blurriness typically observed in VAEs, and expand on generalization to longer sequences.